

# Water Uptake by Fresh Indonesian Peat Burning Particles is Limited by Water Soluble Organic Matter

**Jing Chen[1, *], Sri Hapsari Budisulistiorini[1], Masayuki Itoh[2], Wen-Chien Lee[1, 3], Takuma Miyakawa[4], Yuichi Komzaki[4], LiuDongQing Yang[1], and Mikinori Kuwata[1, 2, *]**

[1] {Earth Observatory of Singapore, Nanyang Technological University, Singapore}

[2] {Center for Southeast Asian Studies, Kyoto University, Japan}

[3] {Division of Chemistry and Biological Chemistry, Nanyang Technological University, Singapore}

[4] {Japan Agency for Marine Science and Technology, Japan}

* Correspondence to: chen.jing@ntu.edu.sg; kuwata@ntu.edu.sg





## 1 Abstract

The relationship between hygroscopic properties and chemical characteristics of Indonesian
biomass burning (BB) particles, which are dominantly generated from peatland fires, was
investigated using the humidified tandem differential mobility analyzer. In addition to peat,
acacia (a popular species at plantation) and fern (a pioneering species after disturbance by fire)
were used for experiments. Fresh Indonesian peat burning particles are almost non-hygroscopic
(mean hygroscopicity parameter, $\kappa < 0.06$) due to predominant contribution of water-insoluble
organics. The range of $\kappa$ spans from $\kappa = 0.02-0.04$ (dry diameter = 100 nm, hereinafter) for Riau
peat burning particles, while that for Central Kalimantan ranges from $\kappa = 0.05-0.06$. Fern
combustion particles are more hygroscopic ($\kappa = 0.08$), whereas the acacia burning particles have
a mediate $\kappa$ value (0.04). These results suggest that $\kappa$ is significantly dependent on biomass types.
This variance in $\kappa$ is partially determined by fractions of water soluble organic carbon (WSOC),
as demonstrated by a correlation analysis ($R = 0.65$). $\kappa$ of water soluble organic matter is also
quantified, incorporating the 1-octanol-water partitioning method. $\kappa$ values for the water extracts
are high, especially for peat burning particles ($A_0$ (a whole part of water-soluble fraction): $\kappa =$
0.18, $A_1$ (highly water-soluble fraction): $\kappa = 0.30$). This result stresses the importance of both
WSOC fraction and $\kappa$ of water soluble fraction in determining hygroscopicity of organic aerosol
particles. Values of $\kappa$ correlate positively ($R = 0.89$) with fraction of $m/z$ 44 ion signal quantified
using a mass spectrometric technique, demonstrating the importance of highly oxygenated
organic compounds in controlling hygroscopicity of Indonesian BB particles. These results can
be further utilized for investigating environmental and climatic impacts of Indonesian BB
particles in both regional and global scales.




## 1. Introduction

Tropical peatland fires in Southeast Asia, which occur by combustion of both peat and vegetation on it, have become frequent during the last few decades (van der Werf et al., 2010; Reddington et al., 2014; Marlier et al., 2015; Spracklen et al., 2015; Stockwell et al., 2016). The peatland fires keep smoldering for months, releasing huge amounts of greenhouse gases and fine particles to the atmosphere, impacting the atmospheric radiation (Levine et al., 1999; Page et al., 2002; van der Werf et al., 2010). In addition, the peatland burning particles cause transboundary air pollution, influencing visibility and human health (Kunii et al., 2002; Wang et al., 2004; Marlier et al., 2013; Crippa et al., 2016; Koplitz et al., 2016). These regional and global impacts are closely related with water uptake properties of peatland burning particles, as water vapor alters aerosol physical and chemical characteristics, such as particle diameter.

Water uptake properties of biomass burning particles, including that emitted from peatlands, have been explored in laboratory by hygroscopic growth and cloud condensation nuclei (CCN) measurements (Chand et al., 2005; Dusek et al., 2005; Day et al., 2006; Petters et al., 2009; Carrico et al., 2010). In general, biomass burning particles are hygroscopic. For instance, the reported range of hygroscopicity parameter, $\kappa$, which serves as a metric for water uptake properties, varies from weakly ($\kappa = 0.02$) to highly hygroscopic ($\kappa = 0.80$) for freshly emitted biomass burning particles (Day et al., 2006; Petters and Kreidenweis, 2008; Petters et al., 2009; Carrico et al., 2010). A field observation of water uptake properties of Indonesian biomass burning plumes also demonstrated that these particles are hygroscopic, with a median hygroscopic growth in light scattering ($f$(RH)) of 1.65 between 20% and 80% relative humidity (RH) (Gras et al., 1999). On the other hand, freshly emitted Indonesian peat burning particles generated in a laboratory were suggested to be non-hygroscopic with respect to a quite low $f$(RH) = 1.05 at 90% RH (Chand et al., 2005), and they were almost CCN inactive ($\kappa = 0.05$ for 100 nm particles) (Dusek et al., 2005). The uniqueness of water uptake property of freshly emitted Indonesian peatland burning particles as well as the discrepancy between the previously reported laboratory and field data need to be consistently understood based on their chemical compositions for accurate evaluation on the environmental impacts.



Chemical composition of biomass burning particles, including these from Indonesian
peatland fires, is dominated by a complex mixture of organic species (Jimenez et al., 2009; Ng et
al., 2010; Cubison et al., 2011; Stockwell et al., 2016). The complexity in chemical composition
inhibits understanding their water uptake properties at molecular levels (Asa-Awuku et al., 2008;
Psichoudaki and Pandis, 2013; Riipinen et al., 2015). To overcome this difficulty, classification
of organic compounds using multiple solvents (Carrico et al., 2008; Polidori et al., 2008; Chen et
al., 2016), liquid-liquid extraction using 1-octanol and water (Kuwata and Lee, 2017), and solid
phase extraction (Asa-Awuku et al., 2008) has been conducted. Functional group analysis of
segregated organic matter has also been demonstrated as a strong tool to characterize complex
mixture of organic compounds (Chen et al., 2016). For instance, chemical characteristics of
water soluble organic matter (WSOM) have been intensively investigated, revealing that WSOM
is predominantly consisted of levoglucosan-like species, carboxylic acids, aldehydes, ketones,
aliphatic alcohols, and polyacids (Decesari et al., 2000; Peng et al., 2001; Suzuki et al., 2001;
Mayol-Bracero et al., 2002; Chan et al., 2005; Psichoudaki and Pandis, 2013). Recently, the
important roles of functional groups on water uptake properties were also investigated by both
theoretical and experimental approaches (Suda et al., 2014; Petters et al., 2016).
In this study, hygroscopic growth of Indonesian peatland burning particles was
investigated by a series of laboratory experiments to understand the relationships between water
uptake properties and chemical characteristics. Hygroscopic growth of fresh peat/biomass
burning particles was measured using the humidified tandem differential mobility analyzer
(HTDMA) (Massling et al., 2003; Gysel et al., 2004; Carrico et al., 2008, 2010; Dusek et al.,
2011). Chemical characterization was also conducted using the Aerodyne Time of Flight-Aerosol
Chemical Speciation Monitor (ToF-ACSM). In addition, ratios of water soluble organic carbon
(WSOC) to organic carbon (OC) were quantified. These measurements were also conducted for
WSOM fraction. Furthermore, WSOM was fractionated by the 1-octanol-water partitioning
method, providing data on hydrophilic fraction (Kuwata and Lee, 2017). These data were
synergistically combined to provide a detailed picture on water uptake properties of freshly
emitted Indonesian peatland burning particles.
**2. Experiment**





## 2.1. Combustion setup

Figure 1(a) shows the experimental setup. Peat and biomass samples were collected at peatlands in Riau and Central Kalimantan provinces in Indonesia (Table 1). The sampling sites include both burnt and undisturbed forest areas. In this region, peatland fire frequently reoccurs, and vast areas are experiencing regeneration of vegetation after fire events. The peat samples were segregated for different sampling depths, as detailed in Table 1. Two other types of biomasses from Riau, including *pteridium aquilinum* (called as fern here) and leaves of *acacia mangium* (abbreviated as acacia), were also employed for the experiment. Fern is one of the major pioneer species after peatland fires (Aswin et al., 2004). Acacia is one of the representative trees for plantations over drained peatland. Both acacia and fern samples were dried at ambient temperature after sampling. Further detailed information on the biomass samples are available in Budisulistiorini et al. (2017).

The biomass samples were used without desiccation. Approximately 1.0 g of biomass sample was combusted in a sealed 100 L stainless steel container using a crucible, which was heated at 350 °C by a ribbon heater, thermocouple, and PID controller (Kuwata et al., 2017). The target heating temperature was normally achieved within 2 − 3 min. Visual inspection confirmed that the combustion condition was mostly smoldering, consistent with a previous report (Usup et al., 2004). Particle-free air was continuously supplied to the container. Excess amount of particle-free air was released to the laboratory, allowing conducting the experiments at room pressure. Particles generated by the burning experiments were diluted by a two-stage dilution system. Size distributions of biomass burning particles were measured using the Scanning Mobility Particle Sizer (SMPS, TSI Inc.). The measurement range of the SMPS was set as 14.6 − 685.4 nm, and time resolution was 3-min. Chemical compositions of particles were quantified using the Aerodyne ToF-ACSM (Fröhlich et al., 2013), while water uptake property was measured using the HTDMA (Massling et al., 2003, 2007; Duplissy et al., 2009). Online measurements with SMPS, ToF-ACSM, and HTDMA were all operated following the dilution. Each combustion experiment lasted for ~ 1 h. Detailed descriptions about the ToF-ACSM and HTDMA are provided in the following sections.

Two filter samples were also collected simultaneously for each of the experiments. Particles were collected onto two 47 mm diameter filters housed in stainless steel filter holders





(BGI Inc.) for half an hour at flowrates of 5.0 lpm. Teflon filters (0.2 μm pore size, Fluoropore™,
Sigma Aldrich) were used for WSOM samples, while quartz filter samples were employed for
carbon analysis by the thermal-optical method. The collected samples were stored under
refrigeration at −20 °C until analysis.
**2.2. Extraction and nebulization of WSOM**

6       Filter samples were extracted using approximately 20 ml of ultrapure water (Type I) by

sonicating them for 30-min at room temperature. The resulting solutions were filtered through
0.2 μm PTFE syringe filters (514-0070, VWR), yielding water extracts (denoted as $A_0$, i.e., a
whole part of or bulk water-soluble fraction). An aliquot of $A_0$ was mixed with the same volume
(5 ml) of 1-octanol (Wako first grade, Wako) using a separatory funnel for classification by 1-
octanol-water extraction (Valvani et al., 1981). The aqueous phase following the 1-octanol-water
extraction is denoted as $A_1$ (slightly less than 5 ml), corresponding to the highly water-soluble
fraction. Details of the extraction method are provided in Kuwata and Lee (2017).

14       The aqueous solutions were nebulized using a glass nebulizer. A mass flow controller

(MC-20 SLPM-D, Alicat Scientific, Inc.) was used to regulate the flow rate (3.5 lpm) of particle-
free air supplied to the nebulizer. Following nebulization, the sample was desiccated by a
diffusion dryer (Model 42000, Brechtel Manufacturing, Inc.) filled with silica gel (Chameleon
83000.360, VWR International). The desiccated particles were measured using the HTDMA,
ToF-ACSM, and SMPS. The analysis of $A_0$ and $A_1$ were conducted only for peat (sampled from a
burnt area, Riau-4), acacia, and fern samples.
**2.3. HTDMA**

22       Hygroscopic growth of particles was measured using the HTDMA (Massling et al., 2003,

2007; Duplissy et al., 2009; Gysel et al., 2009). The HTDMA system consists of three major
components: 1) the first DMA (TSI Inc. Model 3081) to select monodisperse particles of a
specific diameter, 2) the humidification unit for hydrating the classified particles at a target RH,
and 3) the second DMA (TSI Inc. Model 3081) and a condensation particle counter (CPC, TSI
Inc. Model 3775) to detect humidified size distributions (Figure S1).



1 Aerosol particles were dried using a diffusion dryer (Model 42000, Brechtel
2 Manufacturing, Inc.), and introduced to the first DMA at a flow rate of 0.3 lpm. The first DMA
3 selected 50, 100, and 200 nm particles. Both the first and second DMAs were operated at sheath-
4 to-sample flow ratios of 10:1. The resulting monodisperse particles were exposed to a predefined
5 RH environment using two Nafion membrane tubes (Permapure Inc. Model MD-110-12S-4) in
6 series. The target RH, which was set at 90%, was regulated by controlling the flow ratio of
7 humidified and dry air flows via a PID controlling software (National Instruments Inc. Labview).
8 The particle residence time between the humidifier and the second DMA was approximately
9 10 seconds. The RH-controlled humid air was used as the sheath flow for the second DMA. The
10 RH and temperature of the second DMA were continuously monitored at both the sample inlet
11 and sheath outlet using two capacitive RH and temperature probes (Rotronic Inc. Hygroclip
12 HC2-S). The RH differences between the sample and sheath flows were less than 2%. RH of
13 humidified sample air was slightly higher than that of the sheath outflow for the setup shown in
14 Figure S1.

15 The operating conditions of the DMAs were checked using $100 \pm 3$ nm polystyrene latex
16 particles (PSL, Thermo Scientific Inc., Cat. No.: 3100A). Hygroscopic growth of particles in the
17 HTDMA was calibrated by measuring growth factor ($g$), which is defined as a diameter ratio of
18 humidified ($D(RH)$) and dry particles ($D_0$) ($g = (D(RH)/D_0)$, of ammonium sulfate particles.
19 The measured value of $g$ at RH = 85% was 1.71 ($D_0$ = 150 nm), which is comparable to a
20 literature data ($g = 1.69$) (Tang and Munkelwitz, 1994). The stability of detected RH of the
21 HTDMA system was within $\pm 1.0\%$ RH (peak-to-peak) for the target RH of 90% (Figure S2).
22 Volume mean growth factor (GF) was calculated using measured probability density function of
23 $g$ ($c(g, D_0)$) as $GF = \left( \int_0^\infty g^3 c(g, D_0) dg \right)^{1/3}$.

24 **2.4. Chemical characterization**

25 The Aerodyne ToF-ACSM was utilized to measure chemical compositions of non-
26 refractory submicron particles (NR-PM$_1$) (Fröhlich et al., 2013). Five specific chemical
27 components, including organic matter (OM), sulfate (SO$_4^{2-}$), nitrate (NO$_3^-$), ammonium (NH$_4^+$),
28 and chloride (Cl$^-$), were quantitatively detected (Allan et al., 2003), with a time resolution of 3-
29 min.



Bulk OC, elemental carbon (EC), and WSOC contents were also analyzed for 10 samples
(Table 3) using quartz-fiber filter samples (47 mm in diameter). All the quartz-fiber filters were
prebaked at 900 ℃ for 3 h before sampling.  The filter samples were stored in a refrigerator (−20
°C) until analysis. For each sampling, a back-up quartz-fiber filter was used to account for
potential influence of adsorption of gas phase organic components (Turpin et al., 1994). OC
loading on the back-up filter was subtracted from that on the front filter to estimate particulate
OC (i.e., corrected OC).
OC and EC were analyzed by thermal-optical reflectance analysis (Chow et al., 1993)
using a Sunset Laboratory OC/EC Analyzer, following the IMPROVE-A protocol. WSOC was
quantified with a Sievers 800 Total Organic Carbon (TOC) Analyzer after extraction of wildfire
filter samples by water. A portion (8 mm φ) of each WSOC sample was extracted using 10 ml of
HPLC-grade water. The samples were shaken by an orbital shaker for 21 h. The extracted sample
solutions were filtered with syringe filters (pore size of 0.2 μm) prior to injection to the TOC
analyzer. The particulate WSOC of wildfire haze particles was also corrected following the
similar procedure to that of particulate OC.
**3. Results and discussion**
Both the HTDMA and chemical analysis data are summarized in Table 2 and 3. The ToF-
ACSM and OC/EC data (Table 3) demonstrate that chemical composition of submicron biomass
burning particles is dominated by organic species, accounting for approximately 99% in mass
(Budisulistiorini et al., 2017). Contributions of other species, including inorganic ionic species
and EC, were minimal. In the following, the relationships between hygroscopic property and
chemical characteristics of organic species are discussed.
**3.1. Hygroscopic growth factor**
Figure 2 shows normalized particle number size distributions of peat sampled from a
burnt area (Riau-4), acacia leaves, and fern burning particles following hygroscopic growth at
RH = 90% ($D_0$ = 100 nm). The data shown in Figure 2 includes both online (a) and offline ((b):
$A_0$, and (c): $A_1$)) measurements. In all cases, narrow monodisperse distributions were observed,





demonstrating that chemical compositions of particles were uniform (Gysel et al., 2007; Carrico
et al., 2010). For online measurements, diameter change induced by hygroscopic growth was
minimal for the peat and acacia burning particles ($g = 1.05 \sim 1.09$), while diameter of fern
burning particles significantly increased following exposure to high RH ($g = 1.17$). The variation
in hygroscopic properties is originated from differences in organic chemical composition, as
these biomass burning particles contain negligible fractions of inorganic ionic species
(Budisulistiorini et al., 2017).

8        Table 2 summarizes all the values of GF. Values of GF for most of peat samples from

burnt peatland in Riau were less than 1.1. Sampling depths of peat did not significantly affect GF.
There was no clear size-dependence of GF. For instance, GF values of particles from combustion
of peat at drained and burnt areas in Riau were $1.07 \pm 0.04$ ($D_0 = 50$ nm), $1.06 \pm 0.02$ ($D_0 =$
100 nm), and $1.07 \pm 0.02$ ($D_0 = 200$ nm). Particles emitted from the undisturbed forest area in
Riau (i.e., Riau-Zam) were more hygroscopic (GF = 1.11 for $D_0 = 100$ nm) than those generated
from other samples from Riau, while GF of particles emitted by combustion of a peat sample
from the secondary forest in Riau (i.e., Riau-SF) was very similar to those from Riau peat
samples from burnt areas (i.e., Riau-1~4). The similarity is probably due to the short distance
between the two sampling sites (less than 10 km). Particles emitted from peat samples collected
at Central Kalimantan (i.e., C.K.-DB and C.K.-DF) were relatively more hygroscopic (GF > 1.11)
than those from Riau.

20       Hygroscopic growth of bulk water-soluble fraction ($A_0$) is much more significant than

those of the online measurements. Specifically, the mean diameter growth factors were 1.34
(peat sampled from a burnt area, Riau-4), 1.23, (acacia), and 1.28 (fern) for 100 nm particles.
The significant hygroscopic growth of $A_0$ from peat burning particles could be due to high water
uptake by the highly water-soluble fraction, $A_1$ (GF = 1.50). The GFs of $A_1$ for acacia and fern
burning particles were 1.42 and 1.33, respectively. Although water uptake by fresh peat burning
particles was much less than those of vegetation burning particles, the water soluble fraction of
peat burning particles was the most hygroscopic. This result stresses the importance of
understanding hygroscopic properties of WSOM as well as WSOC fraction in total OC.
**3.2. Hygroscopicity parameter ($\kappa$)**





Hygroscopicity parameter ($\kappa$) was calculated using the $\kappa$–Köhler theory (Petters and
Kreidenweis, 2007):

$$\kappa = (GF^3 - 1) \cdot \left( \frac{\exp\left( \dfrac{4\sigma_{s/a} \cdot M_w}{R \cdot T \cdot D_0 \cdot GF} \right)}{RH} - 1 \right) \tag{1},$$

where $\sigma_{s/a}$ is the surface tension of the solution/air interface (0.0718 N m$^{-1}$ at 25 °C), $M_w$ is the
molecular weight of water (18 g mol$^{-1}$), $R$ is the universal gas constant (8.31 J K$^{-1}$ mol$^{-1}$) and $T$ is
temperature (298 K). The $\kappa$ value is related with molar volume of water soluble compounds
($M_s/\rho_s$), which is calculated from both the molecular weight ($M_s$) and density ($\rho_s$) by the
following equation (Rose et al., 2008).

$$\kappa = i_s \frac{\rho_s M_w}{\rho_w M_s} \tag{2},$$

where $i$ is van't Hoff factor, $\rho_w$ is the density of water. The calculated values of $\kappa$ are
summarized in Figure 3 and Table 2.
The range of $\kappa$ for peat burning particles in Riau (sampled from burnt areas) is 0.02 to
0.04, while that for Central Kalimantan samples is 0.05 to 0.06 (100 nm). These values may be
compared with CCN activity of peat burning particles reported by Dusek et al. (2005). Based on
the experimental data by Dusek et al. (2005), the critical supersaturation for CCN activation of
Indonesian peat burning particles is derived as 0.53% for 100 nm particles. This value can be
converted to $\kappa$ of 0.05, which is very similar to the values summarized in Figure 3 and Table 2.
The consistently low values of $\kappa$ suggest that water uptake by freshly emitted peat burning
particles is minimal. The range of $\kappa$ observed for acacia and fern burning particles ($\kappa = 0.04-0.08$)
is comparable to that observed for less hygroscopic mode by previous laboratory experiments on
biomass burning particles (Carrico et al., 2010).
The values of $\kappa$ observed for water extracts ($A_0$) span from 0.11 (acacia, 100 nm) to 0.18
(peat sampled from a burnt area, Riau-4, 100 nm) (Figure 4 and Table 2). The $\kappa$ value for peat
burning particles ($A_0$) is significantly higher than those emitted from the acacia and fern leaves,





highlighting the importance of understanding hygroscopicity of WSOM as well as water soluble
fraction in quantitatively understanding water uptake properties. The value of $\kappa$ for acacia
burning particles is similar to that was measured for WSOM extracted from a prescribed forest
fire experiment in Georgia (USA) ($\kappa = 0.10$), which was estimated from a molar volume of $1.6 \times$
$10^{-4}$ $m^3$ $mol^{-1}$ (Asa-Awuku et al., 2008).

6         The $\kappa$ values for $A_1$ are higher than those for $A_0$. Namely, $\kappa$ observed for $A_1$ were 0.30

(peat sampled from a burnt area, Riau-4), 0.24 (acacia), and 0.18 (fern), respectively. Kuwata
and Lee (2017) demonstrated that classification of WSOM by XAD-8 column, which is one of
the most frequently used materials for solid phase extraction of WSOM, has a strong relationship
with 1-octanol-water partitioning coefficient ($K_{OW}$). Namely, XAD-8 column selectively traps
hydrophobic chemical species in WSOM, which tend to partition to 1-octanol phase ($K_{OW} > 1$).
Thus, hydrophilic fraction separated by XAD-8 is dominantly composed of chemical species,
which has $K_{OW} < 1$. The WSOM in $A_1$ is also dominantly composed of organic compounds with
lower values of $K_{OW}$ ($K_{OW} < 1$), suggesting that $A_1$ and WSOM classified by XAD-8 are
comparable. The $\kappa$ value of biomass burning WSOM separated by XAD-8 is estimated as 0.29,
using molar volume ($6.2 \times 10^{-5}$ $m^3$ $mol^{-1}$) estimated from a CCN measurement by Asa-Awuku et
al. (2008). The comparison provides a typical range of $\kappa$ for hydrophilic ($K_{OW} < 1$) fraction of
WSOM emitted from biomass burning as 0.2~0.3.
**3.3. $\kappa$ (online) and WSOC/OC**

20        WSOC/OC ratios of Indonesian peat and vegetation burning particles are summarized in

Table 3. In general, WSOC/OC ratios for peat burning particles from the burnt area in Riau are
small, ranging from 0.93% to 1.80%. Particles emitted from combustion of peat collected in
other areas tend to contain higher fractions of WSOC (WSOC/OC = 2.03−6.08%). The
variability in WSOC/OC ratio could be due to differences in chemical composition of peat
sampled at different areas (Hikmatullah and Sukarman, 2014). These values are an order of
magnitude lower than the experimental data by Iinuma et al. (2007), which reported WSOC/OC
ratio for Indonesian peat burning particles from South Sumatra as 39%. The significant
difference in WSOC/OC ratio could be originated from the variations in chemical compositions
of peat as well as combustion conditions. Both a systematic laboratory experiment and chemical
analysis of freshly emitted peat burning particles are needed to address the difference in the data.



The WSOC fractions for acacia and fern burning particles were relatively higher (WSOC/OC =
3.42−6.56%) than those from peat combustion.

3            Figure 5 compares $\kappa$ and WSOC/OC ratios. $\kappa$ and WSOC/OC correlate to some extent ($R$

= 0.65), although the variation ranges for both variables are small. Fern burning particles contain
significantly higher fraction of WSOM than other samples, providing an explanation for higher $\kappa$
value for fern burning particles. Nevertheless, the correlation between these two parameters is
not tight. This result suggests that other factors, such as chemical composition and hygroscopic
property of water soluble fraction, should also be considered to quantitatively understand water
uptake property.
**3.4. $\kappa$ and OM mass spectra**

11            Figure 6 shows the ToF-ACSM mass spectra for online, $A_0$, and $A_1$ particles, including

those from peat (sampled at a burnt area, Riau-4), acacia and fern burning. The online mass
spectra have intense signals at $m/z$ 41 ($C_3H_5^+$), 43 (most likely $C_3H_7^+$), 55 ($C_4H_7^+$) and 57 ($C_4H_9^+$),
suggesting that these particles are highly hydrogenated (Canagaratna et al., 2015). On the other
hand, fractions of ions at $m/z$ 44 ($f_{44}$, mostly $CO_2^+$) are limited ($f_{44} < 0.02$), especially for peat
burning particles. This result signifies that the freshly emitted Indonesian biomass burning
particles, especially those from peat, are not highly oxygenated (Ng et al., 2011). This is in
accordance with a previous study, which showed that $f_{44}$ values for primary hydrocarbon-like
organic compounds are usually less than 0.05 (Ng et al., 2011). In addition, $m/z$ 60 and $m/z$ 73
(mainly from $C_2H_4O_2^+$ and $C_3H_5O_2^+$, respectively), marker ions of levoglucosan-like species (a
tracer for cellulose in biomass burning particles) (Simoneit et al., 1999; Cubison et al., 2011),
were especially pronounced for fern burning particles. These results are also supported by
functional group analysis by proton nuclear magnetic resonance ([1]H NMR) technique, which
contains strong signals originated from levoglucosan-like species (Lee et al., in preparation).

25            The mass spectra of $A_0$ are significantly different from those of online measurements. The

most abundant ion in the mass spectra of $A_0$ is $m/z$ 44. Hydrocarbon peaks, such as $m/z$ 41, 43, 55,
and 57, are still significant, yet less abundant than those of the online measurements. In addition,
contributions of $m/z$ 60 and 73 are also enhanced. These results consistently support the idea that



$A_0$ fraction is highly oxygenated. Especially, $A_0$ fraction for peat burning particles is much more
oxidized than those of fern and acacia samples.

3        The mass spectra of $A_1$ from acacia and fern burning show that the $A_1$ fraction is more

oxidized than $A_0$, as indicated by higher values of $f_{44}$. For instance, $f_{44}$ of $A_1$ from fern burning
particles is 0.08, while that of $A_0$ is 0.05. Another notable characteristic of $A_1$ mass spectra is the
smaller fraction of high molecular weight (HMW) ions, which is observed for the region of $m/z >$
100. The HMW fractions ($f_{HMW}$) for $A_0$ and $A_1$ are 15.8% and 16.0% (peat sampled at a burnt area,
Riau-4), 21.8% and 11.6% (acacia), and 17.4% and 8.2% (fern), which are significantly lower
than the corresponding values for online measurements (Table 3). These results suggest that $A_1$
contains less fractions of high molecular weight species, although decomposition during
ionization process makes the estimation of actual contributions of these compounds difficult.

12        Figure 7 displays $f_{44}$, $f_{60}$, and mean $\kappa$ for different types of Indonesian biomass burning

particles. The data points in Figure 7 distribute to two different regions. Low $f_{44}$ and $f_{60}$ values are
observed for particles emitted from Sumatran peat burning (i.e., Riau-1, -2, -3, -4). The $f_{44}$ of
acacia burning particles is slightly higher but $f_{60}$ is low. By contrast, distinctly higher $f_{44}$ and $f_{60}$
are observed for fern, undisturbed peat (Riau), and peat (Central Kalimantan) burning particles.
This is also in accordance with [1]H NMR analysis, which suggests that functional group
distributions of peat and acacia burning particles are significantly different from that emitted
from fern burning particles (Figure S3). The higher WSOC fraction and $\kappa$ of fern burning
particles could be related with the higher $f_{44}$ and $f_{60}$.

21        Figure 8 correlates $\kappa$ with $f_{44}$ for both online and offline measurements of peat sampled at

a burnt area (Riau-4), acacia and fern burning particles. Similar correlations for all the biomass
burning samples are displayed in Figure S4 of the supplementary material. The correlation of
these two variables is represented as $\kappa = 2.31 \times f_{44} + 0.02$ ($R = 0.89$). The slope is very close to
that reported for the relationship between the hygroscopicity of organics ($\kappa_{org}$) and $f_{44}$ ($\kappa_{org} =$
$2.2 \times f_{44} - 0.13$ , Duplissy et al., 2011). The correlation demonstrates that the degree of
oxidation, which is represented by $f_{44}$, is the key controlling parameter in determining
hygroscopicity of Indonesian biomass burning particles. As discussed above, $f_{44}$ of peat burning
particles (Riau-4) is extremely small (Table 3). Minimal water uptake by peat burning particles





(Riau-4) could be associated with low water soluble fraction (Table 3), considering that both $\kappa$
and $f_{44}$ for the corresponding $A_0$ or $A_1$ sample are high.

3        The correlation shown in Figure 8 has a significant divergence, especially at the region

for high $f_{44}$ and $\kappa$ values (upper right corner of the figure), suggesting that degree of oxidation is
not the only one parameter, which controls water uptake property. Especially, comparison of $A_0$
and $A_1$ for peat burning particles highlights the limitation of correlating $f_{44}$ and $\kappa$. The values of
$f_{44}$ for these two fractions are almost the same, while $\kappa$ for $A_1$ is significantly higher than that for
$A_0$. As discussed in section 3.2, the difference of $A_0$ and $A_1$ could be related to that of
hydrophobic and hydrophilic WSOM separated by XAD-8 (Graber and Rudich, 2006; Sullivan
and Weber, 2006). The hydrophobic fraction separated by XAD-8 is typically considered as
humic-like substances (HULIS), which has high molecular weight (e.g., fulvic and humic acids,
Gysel et al., 2004; Graber and Rudich, 2006; Fan et al., 2013). These results suggest that
quantification of HULIS as well as evaluation of their water uptake property will be important
for understanding hygroscopicity of biomass burning particles, including those emitted from
Indonesian peatland fires.

## 17    4. Conclusions

18        Hygroscopic growth of freshly emitted Indonesian biomass burning particles was

investigated in laboratory using the humidified tandem differential mobility analyzer (RH =
90%). The biomass samples included peat, fern, and acacia leaves collected at Riau and Central
Kalimantan in Indonesia. Hygroscopicity was measured for the freshly emitted particles (online),
bulk water-soluble fraction ($A_0$), and highly water-soluble fraction (i.e., fraction with lower $K_{OW}$
values) classified by the 1-octanol-water partitioning method ($A_1$). Hygroscopicity parameter $\kappa$
was derived from the growth factor data. Chemical compositions of these particles were also
quantified using both online and offline techniques.

26        Hygroscopicity of fresh Indonesian peat burning particles is highly dependent on the

origin and condition (e.g., pristine and disturbed) of peat. Particles emitted from combustion of
disturbed peat in Riau were nearly non-hygroscopic (mean $\kappa$ = 0.02−0.04), while those from



undisturbed areas were more hygroscopic ($\kappa$ = 0.03−0.05). Particles emitted from Central
Kalimantan peat samples were generally more hygroscopic ($\kappa$ = 0.05−0.06) than those from Riau.
For biomass samples, acacia burning particles were slightly hygroscopic ($\kappa$ = 0.03−0.04), and
fern burning particles were the most hygroscopic ($\kappa$ = 0.04−0.09) among all samples. These
values loosely correlated with ratio of WSOC to OC ($R$ = 0.65). This result suggests that WSOC
fraction could play an important role in determining $\kappa$, yet other factors, such as difference in
hygroscopicity of slightly and highly water-soluble fractions, should also be considered.

8         Hygroscopicity data of $A_0$ and $A_1$ fractions were significantly different from those of

online measurements. The values of $\kappa$ for $A_0$ were 0.17−0.20 (Riau disturbed peat), 0.09−0.14
(acacia), and 0.10−0.16 (fern). These values were even higher for $A_1$ ($\kappa$ = 0.26−0.31 for Riau
disturbed peat, $\kappa$ = 0.19−0.24 for acacia, and $\kappa$ = 0.16−0.20 for fern). These results demonstrate
that the low hygroscopicity of Riau peat burning particles (online) is due to small water soluble
fraction.

14         The variation in $\kappa$ was related with aerosol mass spectra of organics. $f_{44}$, which is an

indicator for degree of oxygenation, correlated well with $\kappa$ ($R$ = 0.89), demonstrating that
oxygenated functional groups are important for water uptake. In addition, comparison of $A_0$ and
$A_1$ data suggested the importance of high molecular weight species, such as humic-like
substances, in determining the magnitude of hygroscopicity for water soluble fraction.

19         Our experimental results are consistent with previous laboratory studies, which reported

insignificant water uptake by fresh Indonesian peat burning particle (Chand et al., 2005; Dusek et
al, 2005). On the contrary, Gras et al. (1999) showed that particles observed in wildfire plume
from Kalimantan were hygroscopic. The differences between field observation and laboratory
experiments could be caused by atmospheric processes (e.g., secondary formation and chemical
aging of particles), and likely resulted from the differences of burnt materials, their origins and
combustion conditions in practical situations as well. In the future, observations of both chemical
composition and hygroscopic growth of particles emitted from peatland fires need to be
conducted both at vicinity and downstream regions to address the question. The last but not the
least, results obtained from this work can be further developed and applied for modeling studies,

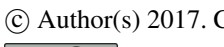



improving estimation of aerosol radiative forcing introduced by Indonesian peat burning
particles in both regional and global scales.
**Acknowledgements**
This work was supported by the Singapore National Research Foundation (NRF) under its
Singapore National Research Fellowship scheme (National Research Fellow Award,
NRF2012NRF-NRFF001-031), the Earth Observatory of Singapore, and Nanyang Technological
University. M. I. was funded by the Ministry of Education, Culture, Sports, Science, and
Technology for Science Research (15H05625), the Ministry of Environment for Global
Environment Research (4-1504), and Research Institute for Humanity and
Nature (RIHN), Japan. T. M. and Y. K. were supported by the Environment Research and
Technology Development Fund (2-1403) of the Ministry of Environment,
Japan, and the Japan Society for the Promotion of Science (JSPS),
KAKENHI Grant number JP26550021. We acknowledge the help of Harris Gunawan and
Satomi Shiodera in some biomass samples collection in Indonesia. We also thank Gissella B.
Lebron for assisting in the laboratory combustion experiments and Pavel Adamek for improving
the English writing.



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



**Table 1.** Summary of Indonesian peat and biomasses used for the laboratory combustion
experiments. Samples 9, 15, and 16 were also used for offline experiments of their aqueous
extracts ($A_0$ and $A_1$).

| Exp. No. | Sample name | Sample depth (cm) | Type /Species | Location | Description |
|---|---|---|---|---|---|
| 1 | Riau-1 | Surface ~ 10 | Peat | Riau, Sumatra | D&B[*] peatland |
| 2 | Riau-1.1 | 10−20 | Peat | Riau, Sumatra | D&B[*] peatland |
| 3 | Riau-1.2 | 30−40 | Peat | Riau, Sumatra | D&B[*] peatland |
| 4 | Riau-2 | Surface ~ 10 | Peat | Riau, Sumatra | D&B[*] peatland |
| 5 | Riau-2.1 | 10−20 | Peat | Riau, Sumatra | D&B[*] peatland |
| 6 | Riau-2.2 | 30−40 | Peat | Riau, Sumatra | D&B[*] peatland |
| 7 | Riau-3 | Surface ~ 10 | Peat | Riau, Sumatra | D&B[*] peatland |
| 8 | Riau-3.1 | 10−20 | Peat | Riau, Sumatra | D&B[*] peatland |
| 9 | **Riau-4[**]** | **Surface ~ 10** | **Peat** | **Riau, Sumatra** | **D&B[*] peatland** |
| 10 | Riau-4.1 | 10−20 | Peat | Riau, Sumatra | D&B[*] peatland |
| 11 | Riau-SF | Surface ~ 5 | Peat | Riau, Sumatra | Secondary forest |
| 12 | Riau-Zam | Surface ~ 5 | Peat | Riau, Sumatra | Undisturbed peat forest |
| 13 | C.K.-DF | Surface ~ 5 | Peat | Palangkaraya, Central Kalimantan | D&UB[#] peat forest |
| 14 | C.K.-DB | Surface ~ 5 | Peat | Palangkaraya, Central Kalimantan | D&B[*] peat forest |
| 15 | **acacia[**]** | **N/A** | **Acacia mangium** | **Riau, Sumatra** | **Dried leaves** |
| 16 | **fern[**]** | **N/A** | **Pteridium aquilinum** | **Riau, Sumatra** | **Dried leaves** |

* D&B stands for the drained and burnt condition.
# D&UB represents the drained but unburnt case.
** Teflon filter samples were collected during online combustion experiments.





1 **Table 2.** Volume weighted mean GF and $\kappa$ values (average ± 1 standard deviation, S.D.) of

2 different types of Indonesian peat and biomasses. The results of their aqueous extracts ($A_0$ and

3 $A_1$) are also shown.

| | Sample name | | Mean GF (RH = 90%) | | | Mean $\kappa$ (RH = 90%) | | |
|---|---|---|---|---|---|---|---|---|
| | | | 50 nm | 100 nm | 200 nm | 50 nm | 100 nm | 200 nm |
| **Online** | **Sumatra** | Riau-1 | 1.17 ± 0.07 | 1.09 ± 0.06 | 1.04 ± 0.01 | 0.089 ± 0.042 | 0.039 ± 0.028 | 0.016 ± 0.002 |
| | | Riau-1.1 | 1.15 ± 0.06 | 1.05 ± 0.01 | 1.04 ± 0.01 | 0.080 ± 0.036 | 0.021 ± 0.005 | 0.014 ± 0.002 |
| | | Riau-1.2 | 1.00 ± 0.002 | 1.08 ± 0.04 | 1.06 ± 0.01 | 0.001 ± 0.001 | 0.036 ± 0.017 | 0.025 ± 0.002 |
| | | Riau-2 | – | 1.07 ± 0.01 | 1.12 ± 0.06 | – | 0.029 ± 0.005 | 0.052 ± 0.032 |
| | | Riau-2.1 | 1.06 ± 0.04 | 1.06 ± 0.01 | 1.06 ± 0.01 | 0.029 ± 0.019 | 0.023 ± 0.005 | 0.024 ± 0.005 |
| | | Riau-2.2 | 1.05 ± 0.02 | 1.09 ± 0.02 | 1.10 ± 0.02 | 0.024 ± 0.010 | 0.037 ± 0.009 | 0.042 ± 0.009 |
| | | Riau-3 | 1.07 ± 0.04 | 1.05 ± 0.01 | 1.08 ± 0.01 | 0.035 ± 0.022 | 0.022 ± 0.006 | 0.033 ± 0.006 |
| | | Riau-3.1 | 1.05 ± 0.02 | 1.05 ± 0.02 | 1.08 ± 0.01 | 0.024 ± 0.012 | 0.022 ± 0.010 | 0.031 ± 0.006 |
| | | Riau-4 | 1.04 ± 0.01 | 1.08 ± 0.01 | 1.05 ± 0.002 | 0.017 ± 0.003 | 0.034 ± 0.003 | 0.019 ± 0.001 |
| | | Riau-4.1 | 1.07 ± 0.10 | 0.99 ± 0.01 | 1.02 ± 0.01 | 0.059 ± 0.063 | N/A | 0.007 ± 0.003 |
| | (Burnt peatland) (Secondary forest) (Undisturbed area) | **Riau D&B[#]** | 1.07 ± 0.04 | 1.06 ± 0.02 | 1.07 ± 0.02 | 0.040 ± 0.023 | 0.029 ± 0.010 | 0.026 ± 0.007 |
| | | Riau-SF | 1.04 ± 0.05 | 1.07 ± 0.004 | 1.09 ± 0.01 | 0.025 ± 0.028 | 0.028 ± 0.002 | 0.034 ± 0.003 |
| | | Riau-Zam | 1.10 ± 0.07 | 1.11 ± 0.04 | 1.08 ± 0.004 | 0.053 ± 0.038 | 0.048 ± 0.017 | 0.032 ± 0.002 |
| | **Kalimantan** | C.K.-DF | 1.11 ± 0.06 | 1.13 ± 0.01 | 1.11 ± 0.01 | 0.057 ± 0.033 | 0.058 ± 0.005 | 0.046 ± 0.004 |
| | | C.K.-DB | 1.11 ± 0.05 | 1.12 ± 0.02 | 1.13 ± 0.01 | 0.055 ± 0.028 | 0.054 ± 0.011 | 0.056 ± 0.005 |
| | | acacia | 1.05 ± 0.01 | 1.09 ± 0.01 | 1.09 ± 0.01 | 0.026 ± 0.005 | 0.039 ± 0.006 | 0.037 ± 0.006 |
| | | fern | 1.08 ± 0.02 | 1.17 ± 0.02 | 1.20 ± 0.03 | 0.039 ± 0.011 | 0.078 ± 0.010 | 0.088 ± 0.014 |

**Aqueous extracts: $A_0$ (the water extracts), $A_1$ (the 1-octanol water extracts)**



| | Sample name | | Mean GF (RH = 90%) | | | Mean $\kappa$ (RH = 90%) | | |
|---|---|---|---|---|---|---|---|---|
| | | | 50 nm | 100 nm | 200 nm | 50 nm | 100 nm | 200 nm |
| **Offline** | $A_0$ | **peat**[*] | 1.29 ± 0.05 | 1.34 ± 0.06 | 1.38 ± 0.09 | 0.168 ± 0.039 | 0.179 ± 0.038 | 0.198 ± 0.058 |
| | | acacia | 1.17 ± 0.03 | 1.23 ± 0.04 | 1.28 ± 0.05 | 0.090 ± 0.019 | 0.110 ± 0.025 | 0.135 ± 0.027 |
| | | fern | 1.18 ± 0.03 | 1.28 ± 0.03 | 1.32 ± 0.05 | 0.100 ± 0.016 | 0.141 ± 0.020 | 0.157 ± 0.031 |
| | $A_1$ | **peat**[*] | 1.47 ± 0.06 | 1.50 ± 0.09 | 1.47 ± 0.11 | 0.311 ± 0.052 | 0.302 ± 0.074 | 0.262 ± 0.083 |
| | | acacia | 1.32 ± 0.04 | 1.42 ± 0.03 | 1.44 ± 0.07 | 0.195 ± 0.027 | 0.237 ± 0.023 | 0.239 ± 0.049 |
| | | fern | 1.28 ± 0.04 | 1.33 ± 0.05 | 1.39 ± 0.05 | 0.162 ± 0.026 | 0.177 ± 0.034 | 0.205 ± 0.038 |

− Data is unavailable due to low particle number concentration.
# **Riau D&B** represents the averages of all the D&B peat samples collected from different
depths of the Sumatran peatlands (i.e., samples used for **Exp. 1-10** in Table 1).
* **Peat** in Table 2 refers to the Riau-4 sample collected from burnt peatlands in Sumatra (see
Sect.2 for details).


1  **Table 3.** Summary of chemical characteristics of different types of Indonesian peat and biomass

2  burning particles.

| Sample name | | Mean $\kappa$ (100 nm) | $f_{44}$ (%) | $f_{HMW}$ (%) | OC (mg C) | EC (mg C) | WSOC/OC (%) |
|---|---|---|---|---|---|---|---|
| **Sumatra** | Riau-1 | 0.039 | 0.4 | 42.9 | 12.69 | 0.13 | 0.93 |
| | Riau-2 | 0.029 | 0.7 | 26.9 | 14.08 | 0.12 | 1.80 |
| | Riau-3 | 0.022 | 0.7 | 31.7 | 13.58 | 0.13 | 1.63 |
| | Riau-4 | 0.034 | 0.5 | 29.0 | 18.86 | 0.13 | 1.51 |
| | Riau-SF | 0.028 | 1.7 | 21.1 | 7.64 | 0.07 | 4.15 |
| | Riau-Zam | 0.048 | 1.6 | 23.8 | 2.58 | 0.03 | 6.08 |
| **Kalimantan** | C.K.-DF | 0.058 | 2.0 | 19.5 | 5.58 | 0.05 | 4.16 |
| | C.K.-DB | 0.054 | 1.9 | 19.3 | 7.51 | 0.05 | 2.03 |
| | acacia | 0.039 | 1.1 | 27.0 | 14.61 | 0.05 | 3.42 |
| | fern | 0.078 | 1.9 | 21.5 | 13.34 | 0.07 | 6.56 |
| | **peat**[*] | 0.179 | 9.5 | 15.8 | N/A | N/A | N/A |
| **A0** | acacia | 0.110 | 4.7 | 21.8 | N/A | N/A | N/A |
| | fern | 0.141 | 5.2 | 17.4 | N/A | N/A | N/A |
| | **peat**[*] | 0.302 | 9.3 | 16.0 | N/A | N/A | N/A |
| **A1** | acacia | 0.237 | 6.6 | 11.6 | N/A | N/A | N/A |
| | fern | 0.177 | 7.9 | 8.2 | N/A | N/A | N/A |





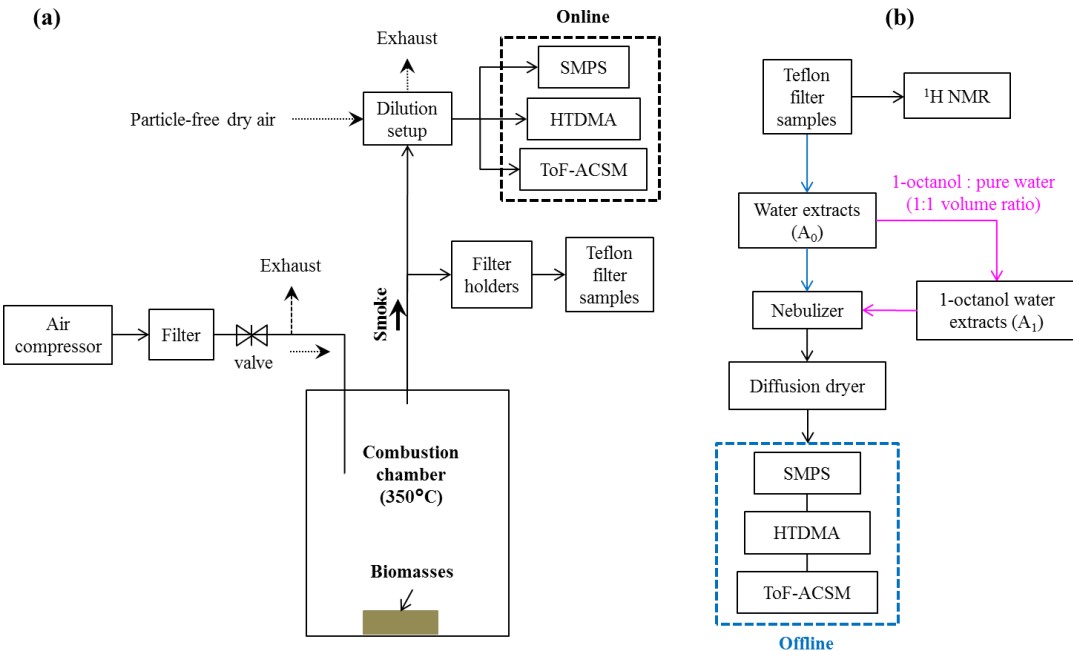

**Fig. 1** Schematic diagrams of the laboratory experimental setups.
(a) The experimental setup for combustion experiment and subsequent online measurements, and
(b) experimental setup for offline measurements of water extracts ($A_0$, blue arrows) and 1-
octanol water extracts ($A_1$, magenta arrows) from the filter samples.



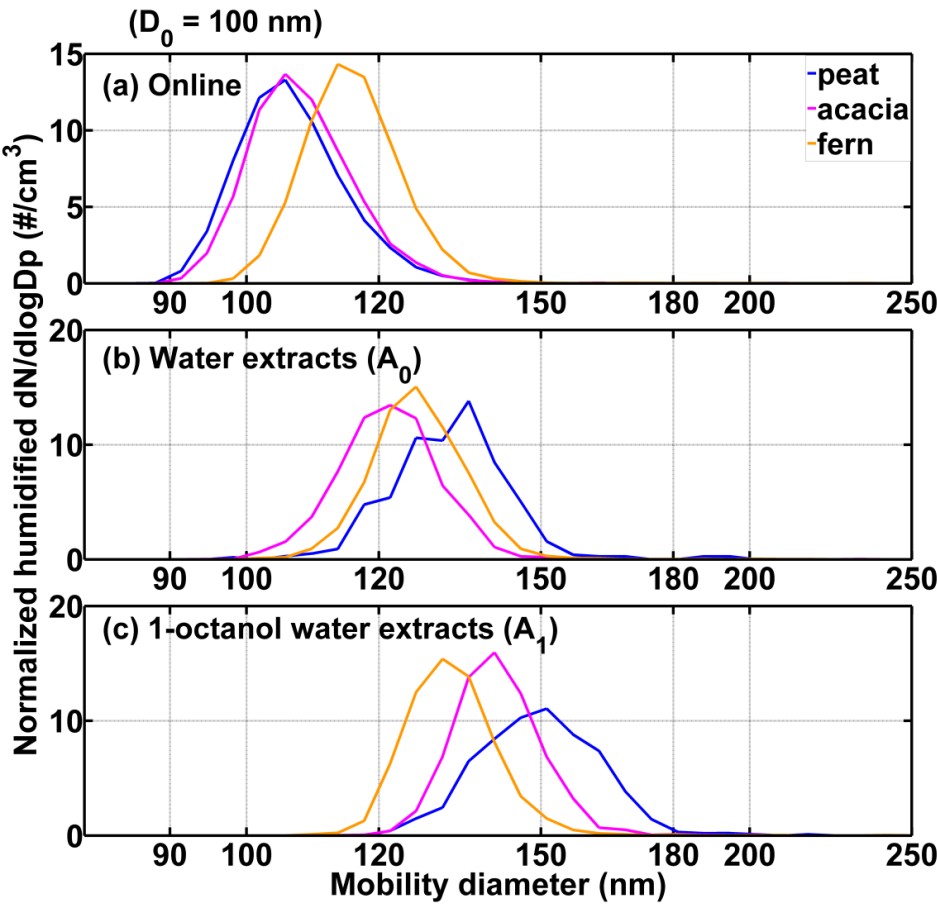

**Fig.2** Normalized number size distributions measured by the HTDMA ($D_0 = 100$ nm; RH = 90%) of peat, fern and acacia burning particles. (a) online data, (b) $A_0$, and (c) $A_1$. For online data, peat burning particles are nearly non-hygroscopic, while fern burning particles are more hygroscopic. Aqueous extracts of peat burning particles are the most hygroscopic among the three types of biomasses. The x-axis is on a logarithmic scale.





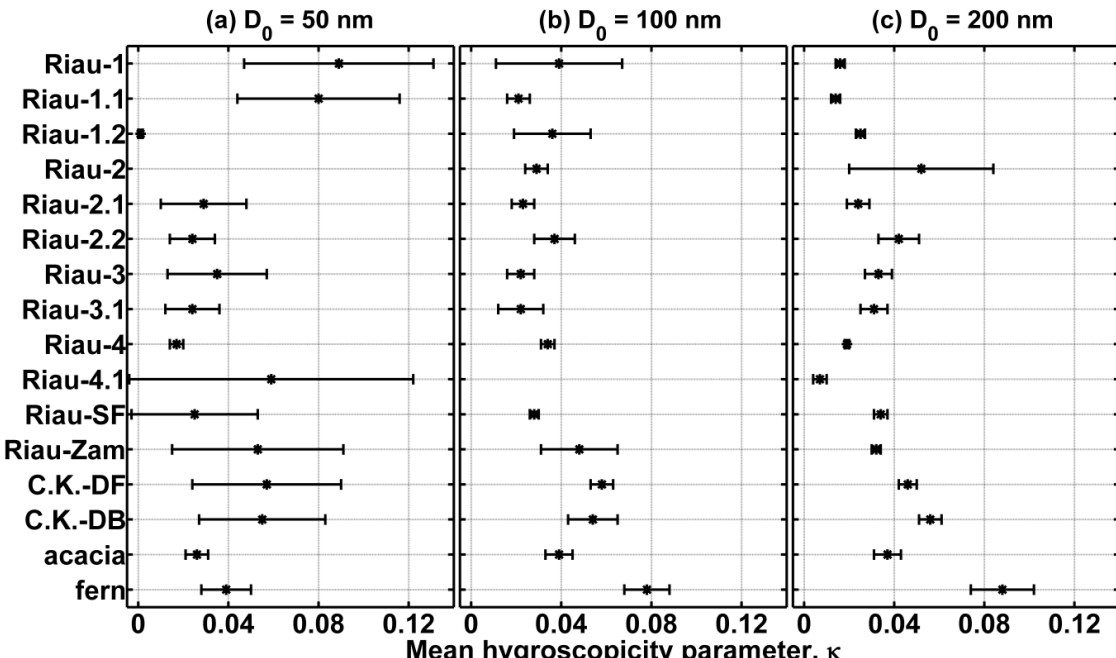

**Fig.3** Mean $\kappa$ values of fresh Indonesian biomass burning particles measured for (a) $D_0$ = 50 nm,
(b) $D_0$ = 100 nm, and (c) $D_0$ = 200 nm particles. In all cases, $\kappa$ values are lower than 0.1. The
largest $\kappa$ values were measured for fern burning particles ($D_0$ = 100 nm and $D_0$ = 200 nm).
Particles emitted from combustion of peat at Central Kalimantan are generally more hygroscopic
than those from Riau, Sumatra. Error bars denote the corresponding standard deviations.





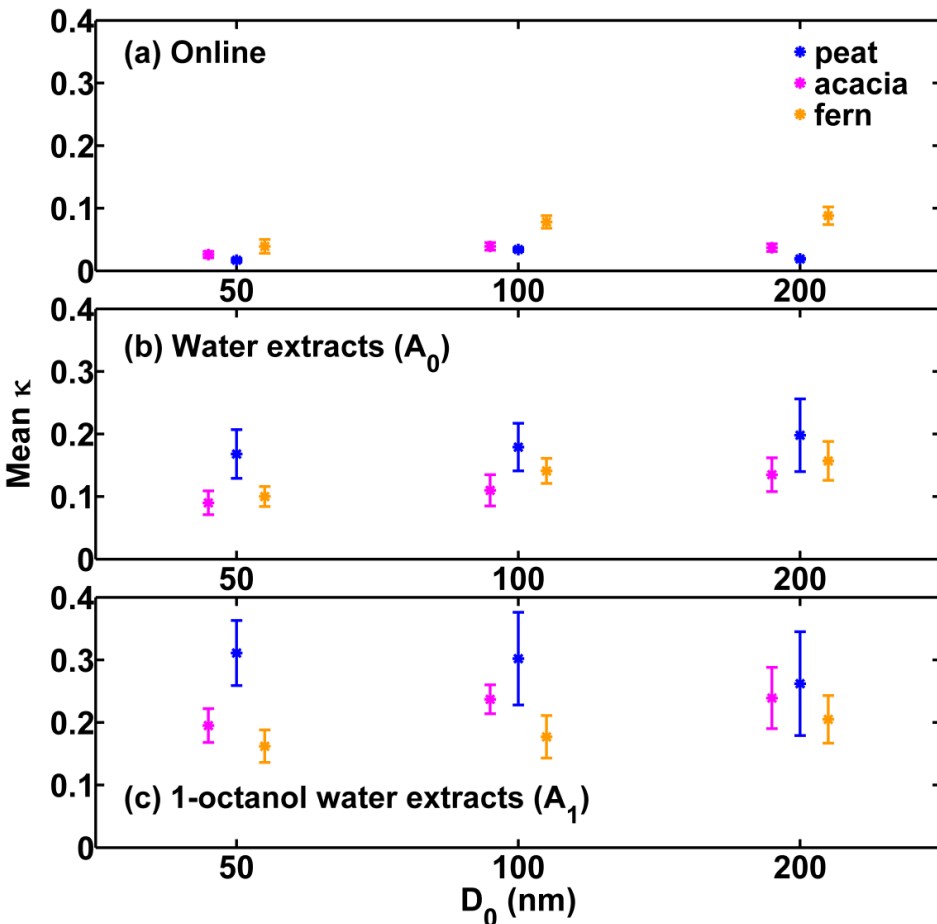

**Fig.4** Comparison of $\kappa$ values for (a) online, (b) water extracts ($A_0$), and (c) 1-octanol water
extracts ($A_1$). The data for peat (Riau-4), acacia, and fern are shown. Indonesian peat burning
particles are almost non-hygroscopic for online data, while their water soluble organic fractions
are highly hygroscopic.



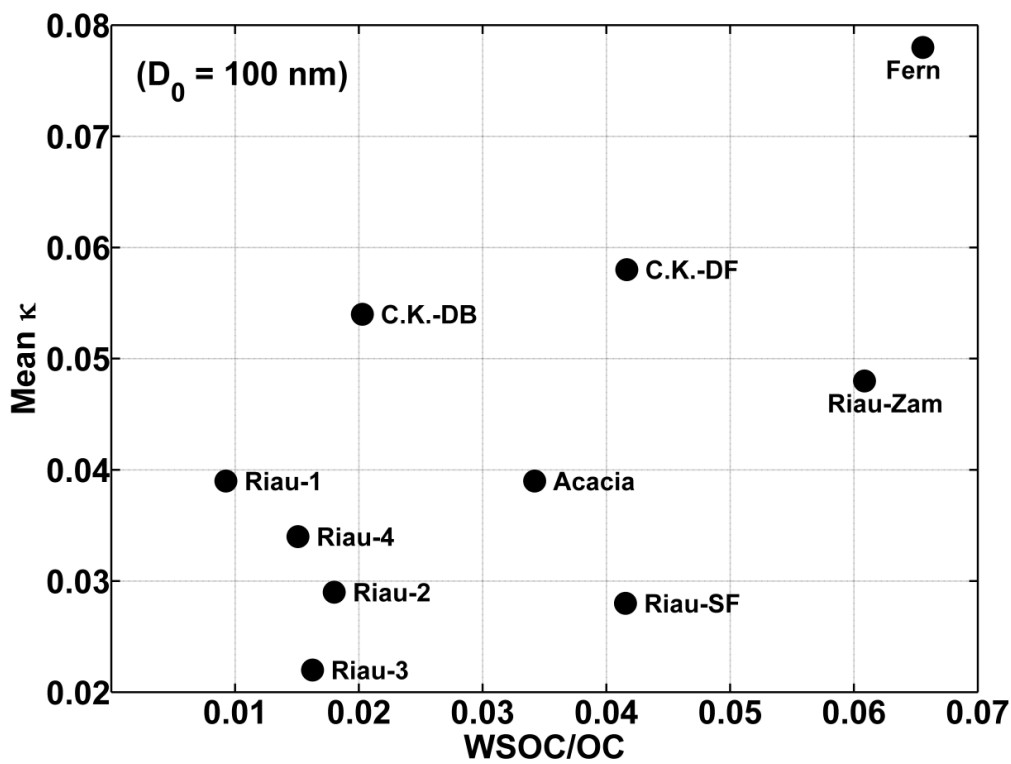

3    **Fig.5** Correlation of $\kappa$ and WSOC/OC ratio for different types of Indonesian peat and biomasses.





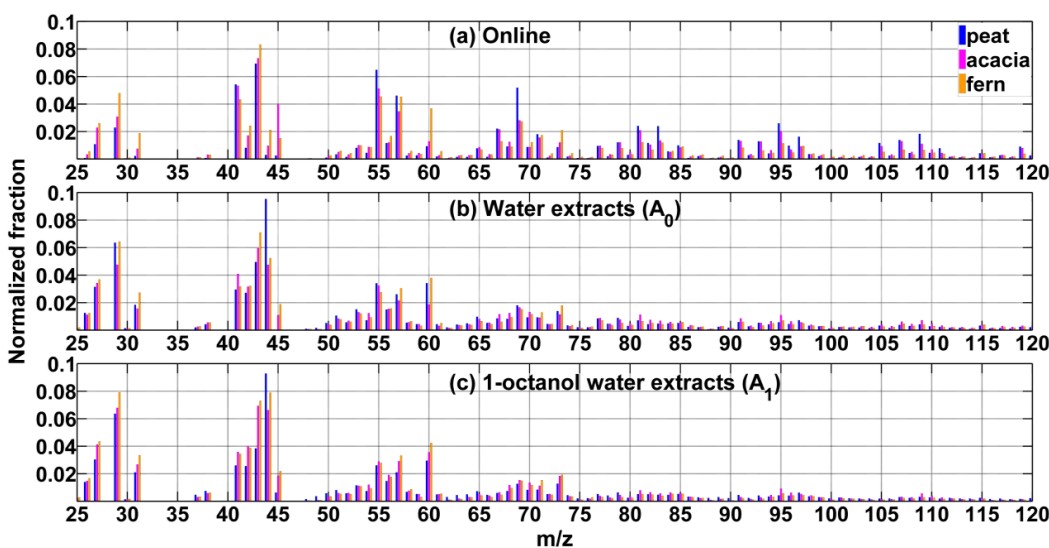

**Fig.6** Mass spectra of organics in Indonesian biomass burning particles measured for peat, acacia,
and fern samples. (a) online, (b) $A_0$, and (c) $A_1$ data are shown. Ion signals ($m/z$) from
hydrocarbon-like organic compounds (e.g., $m/z$ 41, 43, 55, 57) are prominent for online data,
while intensities of ions for oxygenated organics (e.g., $m/z$ 44) and biomass burning tracers (e.g.,
$m/z$ 60, 73) are relatively less intense. On the other hand, both $m/z$ 44 and $m/z$ 60 signals are
significant for mass spectra of $A_0$ and $A_1$. See the text for details.





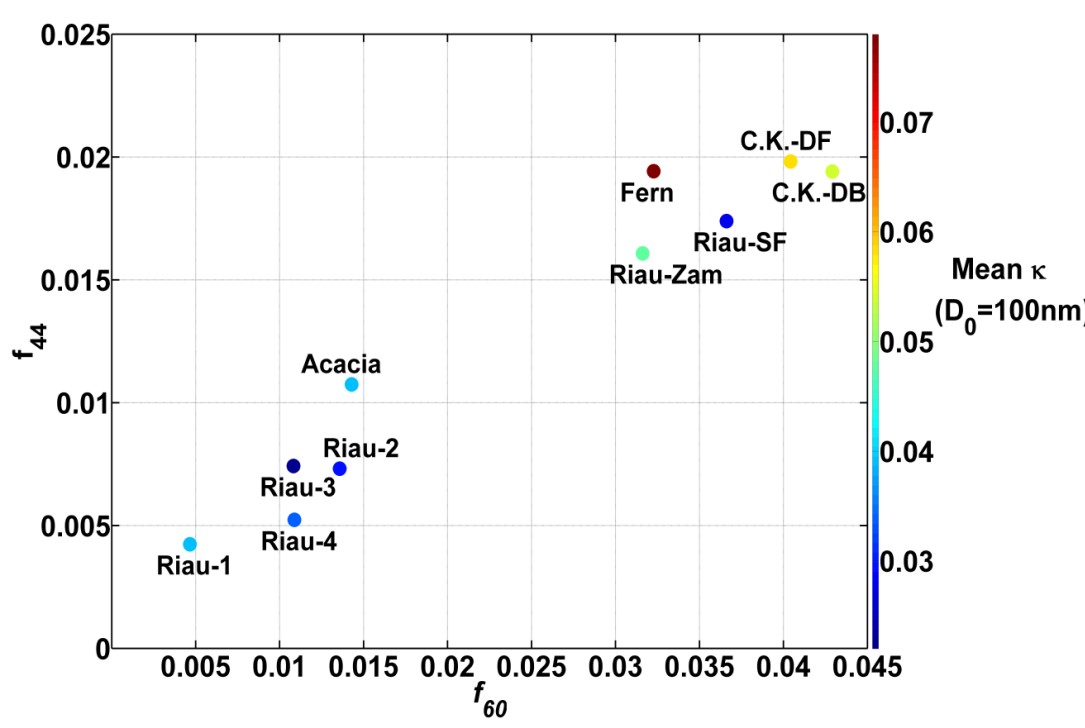

3    **Fig.7** Correlation of $f_{44}$ and $f_{60}$ for different types of Indonesian peat and biomasses.



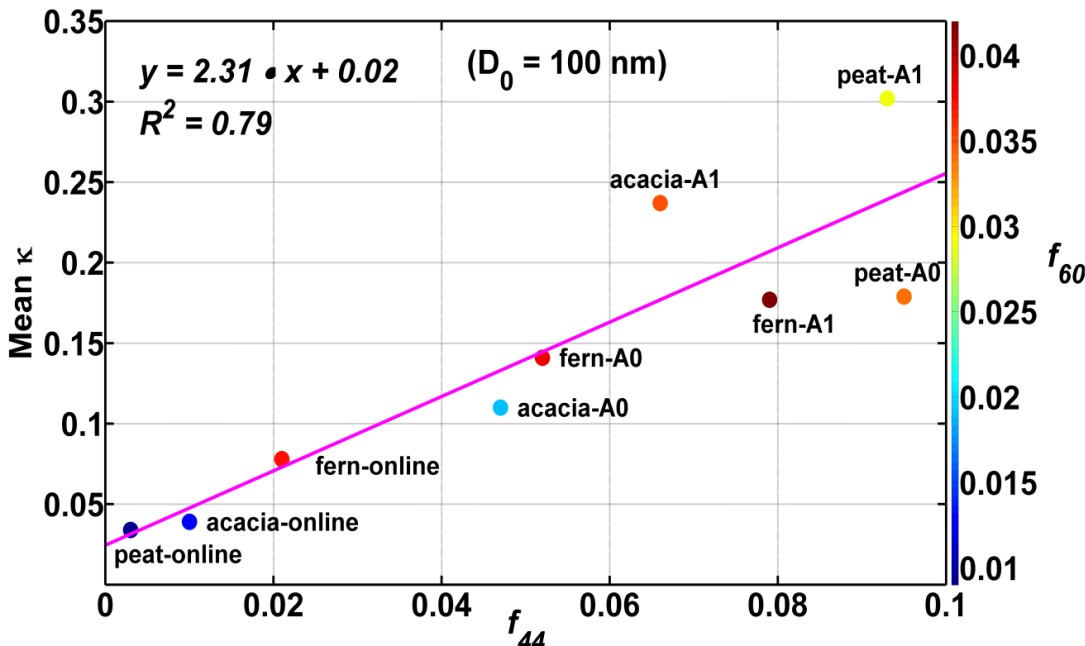

3  **Fig.8** Correlation of $\kappa$ and $f_{44}$. The magenta line shows the result of fitting by the least-squares

4  method.