# Peer review of "Water Uptake by Fresh Indonesian Peat Burning Particles is Limited by Water Soluble Organic Matter"

_Atmospheric Chemistry and Physics, 2017_

## Referee Comment (RC1) · Anonymous Referee #1 · 12 Jun 2017

Chen et al. use the HTDMA technique to study the water uptake properties of biomass burning particles from dried smoldering eagle fern and black wattle plants. Hygroscopic growth was correlated with chemical composition from an ACSM as well as OC/EC ratios, and WSOC content. Smoke collected on filters was extracted using water, and a liquid-liquid ocantol-water extraction. Extracts are atomized and analyzed for chemical composition and hygroscopicity. One of the main findings is a correlation between f44 and the mean kappa observed with the HTDMA.

The manuscript contains new data, collected with a valid set of techniques. The octanol-water extraction method is new. However, the direct utility of the study remains unclear. The authors do not explain the scientific value of analyzing water and octanol-water extracts. How can these be used to help understand biomass burning

aerosol? Yes, products will be different in the extracts, and it may help build correlations of kappa vs f44 over a wider dynamic range, but beyond that, I do not see how the extracts help understanding hygroscopic growth or help improving aerosol forcing estimates as claimed in the conclusion. The authors need to explicitly make this case in the discussion of the results. Furthermore, the correlation of kappa with f44 is now firmly established. Also, the water uptake properties for aerosol from predominately smoldering combustion in small burn settings is also well known to range between 0 and 0.1 from previous studies. Repeating this type of study with more fuels adds only incrementally to the known body of literature. Perhaps the NMR functional group data could be used better for an explanatory model? Overall, the manuscript needs to be revised to demonstrate how the presented data advance the scientific understanding of biomass burning aerosol (what new insight was learned and/or how it could be applied) and then re-reviewed.

Other comments

pg. 8: "In all cases, narrow monodisperse distributions were observed". This has to be evaluated against the width of a truly single component aerosol. The authors should compare the width of the distribution against some standard compound produced by atomization to support their point.

pg. 10: It is unclear why Eq. (2) is provided. The data in the Table 2 appear to be calculated from the data. For a mix of compounds, the Eq. (2) should be formulated for multiple components. Furthermore, Eq. (2) is only valid for infinitely soluble compounds. If the equation is used later, the relationship to solubility should be made clearer, especially in the context to the water and octanol extracts.

pg. 11: "the value of $\kappa$ for acacia burning particles is similar to that was measured for WSOM extracted from a prescribed forest fire experiment in Georgia (USA) ($\kappa = 0.10$), which was estimated from a molar volume of 1.6 $\times$5 10-4 m3 mol -1 (Asa-Awuku et al., 2008)." And "The $\kappa$ value of biomass burning WSOM separated by XAD-8 is estimated as 0.29, using molar volume (6.2 × 10 -5 m 3 mol -1 ) estimated from a CCN measurement by Asa-Awuku et al. (2008)." It is unclear what is meant here. Did Asa-Awuku measure water uptake or CCN and compute kappa? Did they measure (average) molecular weight and kappa is calculated from that? If so, what is the relevance? Please explain.

pg. 12: (Lee et al., in preparation). I believe papers in preparation should not be cited in ACP articles.

───────────────────────────────

---

## Referee Comment (RC2) · Anonymous Referee #2 · 8 Jul 2017

This is a very interesting study focusing on the water uptake of organic-dominated biomass burning aerosols in Indonesia. The very low hygroscopicity was attributed to the small fraction of water soluble organic matter. The authors also demonstrated the importance of biomass types in controlling the kappa and water soluble organic matter fraction and the role of highly oxygenated organic compounds in controlling aerosol hygroscopicity. Overall, it is a convincing study that may help improve our understanding on hygroscopicity of aerosol particles, especially organic aerosols. Previous work was properly referred and the paper was well written. I would recommend its publication if the authors could address a couple of minor issues listed below.

Based on correlations between WSOC fraction and kappa, the authors suggested the importance of WSOC fraction in determining kappa of organic aerosol particles. My

[Figure]

question is if and how we can use the WSOC fraction and kappa of A0 and A1 aerosols to predict kappa, for example under the Zdanovskii, Stokes, and Robin (ZSR) assumption? Do we need additional information to improve the kappa prediction?

Page 3 line 17, can the authors clarify how "hygroscopic" particles or "CCN-active" are defined here? Because a kappa of 0.02 was considered as hygroscopic while a higher kappa of 0.05 was considered as CCN inactive. Concerning the range of kappa in literature, a lower kappa of 0.01 has been reported for low-volatile biomass burning aerosol particles in Rose et al. (2008).

Reference: Rose, D., Gunthe, S. S., Su, H., Garland, R. M., Yang, H., Berghof, M., Cheng, Y. F., Wehner, B., Achtert, P., Nowak, A., Wiedensohler, A., Takegawa, N., Kondo, Y., Hu, M., Zhang, Y., Andreae, M. O., and Pöschl, U.: Cloud condensation nuclei in polluted air and biomass burning smoke near the mega-city Guangzhou, China – Part 2: Size-resolved aerosol chemical composition, diurnal cycles, and externally mixed weakly CCN-active soot particles

---

## Author Comment (AC1) · 3 Aug 2017

The comment was uploaded in the form of a supplement:
https://www.atmos-chem-phys-discuss.net/acp-2017-136/acp-2017-136-AC1-
supplement.zip

———————————————————

---

## Author Comment (AC2) · 3 Aug 2017

The comment was uploaded in the form of a supplement:
https://www.atmos-chem-phys-discuss.net/acp-2017-136/acp-2017-136-AC2-supplement.zip

---

## Author Response (AR1)

Dear Editor,

We would like to thank the two reviewers for their helpful comments and suggestions, which have been fully taken into account upon manuscript revision. A point-by-point response and an accordingly revised manuscript have been uploaded.

In the following, original reviewer comments, our response, and updates on the revised manuscript are shown in **bold**, normal, and *italic*, respectively.

Best Regards,

Jing Chen, Mikinori Kuwata

**Anonymous Referee #1**

**General comments:**

**R1C0: Chen et al. use the HTDMA technique to study the water uptake properties of biomass burning particles from dried smoldering eagle fern and black wattle plants. Hygroscopic growth was correlated with chemical composition from an ACSM as well as OC/EC ratios, and WSOC content. Smoke collected on filters was extracted using water, and a liquid-liquid octanol-water extraction. Extracts are atomized and analyzed for chemical composition and hygroscopicity. One of the main findings is a correlation between $f_{44}$ and the mean kappa observed with the HTDMA.**

**Response:** We appreciate the reviewer for useful comments in revising the manuscript. Our responses to reviewer's concerns are described in the following.

We would like to emphasize that hygroscopic properties of peat smoldering particles were quantified, in addition to those originating from combustion of peatland vegetation (e.g., fern and acacia leaves). To our knowledge, this is the first HTDMA study for particles emitted from smoldering of Indonesian peat, which is one of the most important types of biomass burning fuel. The following sentences were added to the revised manuscript to stress the point.

*Page 3, Line 4: 'As one of the most important biomass burning types, the peatland fires keep smoldering for months, releasing huge amounts of greenhouse gases and fine particles to the atmosphere, impacting atmospheric radiation (Levine et al., 1999; Page et al., 2002; van der Werf et al., 2010).'*

*Page 4, Line 25: 'Hygroscopic growth of various types of fresh peat burning particles, along with those originating from combustion of peatland dried plants,*

*was measured using the humidified tandem differential mobility analyzer (HTDMA) for the first time.'*

**R1C1: "The manuscript contains new data, collected with a valid set of techniques. The octanol-water extraction method is new. However, the direct utility of the study remains unclear. The authors do not explain the scientific value of analyzing water and octanol-water extracts. How can these be used to help understand biomass burning aerosol? Yes, products will be different in the extracts, and it may help build correlations of kappa vs $f_{44}$ over a wider dynamic range, but beyond that, I do not see how the extracts help understanding hygroscopic growth or help improving aerosol forcing estimates as claimed in the conclusion. The authors need to explicitly make this case in the discussion of the results."**

**Response:** We appreciate the reviewer for recognizing the validity of our experimental techniques. The 1-octanol-water partitioning method allows further classification of highly water soluble organic fraction, enabling us to tightly connect chemical composition and hygroscopic properties of biomass burning particles. In previous studies, hygroscopic growth/CCN activity of bulk material, water/methanol extracts, and solid phase extracted organic fractions in biomass burning particles were investigated. Analysis of solvent extracts provided some insights into the relevance between water uptake properties and chemical composition, yet these fractionated components still contained a wide spectrum of organic species in terms of water solubility (e.g., Psichoudaki and Pandis, 2013). Solid phase extractions allowed classification of different fractions in WSOM, yet chemical and physical significances of the separation method have been unclear.

As discussed in our previous study (Kuwata and Lee, 2017), the significance of the 1-octanol-water partitioning method is to classify water soluble organic matter into different fractions by their water solubility, which has a clear chemical meaning. To our knowledge, this study is the first application of the method for hygroscopic growth measurements.

The present study demonstrated for the first time that a highly hydrophilic fraction ($A_1$) is a) more hygroscopic, and b) highly oxygenated, and c) contains a smaller fraction of high-molecular weight species than bulk WSOM ($A_0$). Even though highly oxygenated organic fraction has been considered as highly hygroscopic due to their water solubility, there has been no experimental validation for this idea prior to this study. Therefore, the correlation of $\kappa$ with $f_{44}$ reported in this study provides a clearer idea about the relationship between water uptake properties and organic chemical compositions with different water solubility in comparison to previous studies. In addition, the relationship between molecular size of organic compounds and water solubility has never been demonstrated before, especially for a complex mixture such as biomass burning particles. In conclusion, the significance of the 1-octanol-water partitioning method is more than expanding the dynamic range for the correlation between $\kappa$ and $f_{44}$, as the method adds a new parameter (i.e., water solubility) to characterize chemical properties of WSOM. These points are clarified and stressed in the revised manuscript, as detailed in the following.

*Page 5, Line 1: 'The concurrent HTDMA and ToF-ACSM measurements were also performed for the bulk WSOM and its highly hydrophilic fraction classified with the 1-octanol-water partitioning method in terms of water solubility (Kuwata and Lee, 2017). This method provides a new angle (i.e., water solubility) to characterize chemical properties of WSOM, facilitating a more detailed*

*investigation on particle water uptake property with the first application of the method in HTDMA measurements of highly hydrophilic organic fraction.'*

In previous studies, hygroscopicity/CCN activity has been related with chemical composition only corresponding to the bulk information such as $f_{44}$ or O:C elemental ratios. Although these relationships improved our understanding on interactions between organic aerosol particles and water vapor, it has not been easy to obtain mechanistic insights on the empirical correlations. Recent development of theoretical framework for water uptake by organic aerosol particles suggests that distributions of polarity/water solubility could be an important input for detailed modeling of hygroscopic growth, although experimental verification of the idea is still challenging. The classification of WSOM by their water solubility will allow us to narrow the gap between empirical relationship and theoretical framework based on the first HTDMA measurements of highly water soluble organic fraction. These points are clarified in the revised manuscript as follows.

***Page 12, Line 6:*** *'... biomass burning as 0.2~0.3. Our results of water uptake by organic compounds (e.g., bulk organic material, bulk WSOM, and highly hydrophilic WSOM) would be further employed to verify a theoretical framework, which uses distributions of water solubility as input parameters (Riipinen et al., 2015).'*

**R1C2: "Furthermore, the correlation of kappa with $f_{44}$ is now firmly established. Also, the water uptake properties for aerosol from predominately smoldering combustion in small burn settings is also well known to range**

**between 0 and 0.1 from previous studies. Repeating this type of study with more fuels adds only incrementally to the known body of literature. Perhaps the NMR functional group data could be used better for an explanatory model?***"*

**Response:** We thank the reviewer for the useful comment to improve the manuscript. We would like to stress that our study includes the first HTDMA measurements of bulk organic matter, bulk WSOM, and more hydrophilic WSOM fraction in different types of fresh Indonesian peatland burning particles. As far as we know, no previous HTDMA studies have conducted such intensive investigations on water uptake properties of a broad set of Indonesian peat samples, which possess different origins (e.g., Sumatra and Central Kalimantan) for various depths (i.e., surface ~ 5/10 cm, 10−20 cm, and 30−40 cm) and include different sample conditions (e.g., unburnt and disturbed). Thus, we do not consider that this study is a replication of a firmly established relationship. Rather, this study adds a critical dataset, which will serve as a base to investigate how the peatland fires influence human health, local environment, and the climate. We agree with the reviewer that repetition of existing studies would not have significant impacts. Experimental work on under-investigated yet important phenomena should never be underscored. We have added the following sentences to provide the significance of our study more clearly.

***Page 17, Line 9:*** '*The last but not the least, our results can provide an experimentally validated reference for organics-dominated particle hygroscopicity, thus lowering uncertainties in current climate models and contributing to more accurate estimations of climate impacts caused by Indonesian peatland burning particles in both regional and global scales.*'

We agree with the reviewer that the empirical correlation between $\kappa$ and $f_{44}$ is already known. However, a theoretical background for the empirical relationship is still ambiguous. As stated in our response to R1C1, by relating $\kappa$ to $f_{44}$, this study experimentally evidences the empirical assumption that more oxygenated organics generally correspond to higher water solubility, thus more hygroscopic. Our study suggests a more straightforward chemical composition-dependence, or rather solubility-dependence, of water uptake by organics than previous work. The manuscript was revised as follows to clarify this point.

***Page 14, Line 19:*** '*Figure 8 shows a correlation of $\kappa$ with $f_{44}$ for both online (i.e., bulk organic matter) and offline (i.e., bulk WSOM and highly hydrophilic WSOM fraction) measurements of peat sampled at a burnt area (Riau-4), acacia and fern burning particles.*'

***Page 14, Line 30:*** '*...$f_{44}$ for the corresponding $A_0$ or $A_1$ sample are high. One notable difference of the correlation found in this work from previous studies is the inclusion of highly soluble fraction to the analysis. Although the $\kappa$ - $f_{44}$ correlations have been related to enhanced water solubility, the relationships among these three parameters (i.e., $\kappa$, $f_{44}$, and water solubility) have not been shown prior to this study.*'

The reviewer is right that the $\kappa$ range of 0−0.1 has been commonly reported for water uptake properties of fresh smoldering biomass burning particles over the world. Considering that range of $\kappa$ for pure organic compounds could vary only between 0 and 0.2 (except for oxalic acid), the variability is not negligible (Kuwata et al., 2013). In fact, the difference between $\kappa = 0$ and $\kappa = 0.1$ is physically significant, as the range includes both hydrophobic and generally hygroscopic particles. In our study, we show the corresponding $\kappa$ ranges for three different particle sizes among various types of Indonesian biomasses for the first time. This enriches our understanding of the water uptake by fresh Indonesian peatland burning particles and provides a more reliable reference for climate models with consideration of the hygroscopicity of organics-dominated biomass burning particles especially in the tropical Asian regions.

As the reviewer has pointed out, NMR analysis has a great potential to improve our understanding on how different functional groups contribute to hygroscopic properties. Following descriptions were added to the revised manuscript so that the readers can qualitatively understand how the NMR data could be connected with hygroscopicity measurement.

*Page 14, Line 11: '… fern burning particles. Namely, the peat and acacia samples contain a significantly higher fraction of saturated aliphatic group (i.e., H-C; 71.7 % for peat, and 64.0 % for acacia) in comparison to that in the fern sample (38.6 %, see panel (a) of Figure S4), which readily prohibits the bulk hygroscopic growth of fresh peat burning particles. Besides, the highly polar structure (i.e., H-C-O) in the peat (6.0 %) and acacia (8.1 %) samples is distinctly lower than that in the fern sample (15.5 %, Figure S4(a)). This likely contributes to the higher WSOC fraction of fern burning particles, and the corresponding higher $\kappa$ values could be related with the higher $f_{44}$ and $f_{60}$.'*

**R1C3: "Overall, the manuscript needs to be revised to demonstrate how the presented data advance the scientific understanding of biomass burning**

**aerosol (what new insight was learned and/or how it could be applied) and then re-reviewed."**

**Response:** We acknowledge the reviewer for the critical comments, which are useful in improving the impact and quality of the manuscript. We have fully revised the manuscript according to all the helpful comments and suggestions, and the updated contents can be found as stated above and elsewhere in our responses to the second reviewer's comments.

**Other comments**

**R1C4: pg. 8: "In all cases, narrow monodisperse distributions were observed". This has to be evaluated against the width of a truly single component aerosol. The authors should compare the width of the distribution against some standard compound produced by atomization to support their point.**

**Response:** We thank the reviewer for the useful comment. We have added the comparison of mean normalized particle number size distributions among five types of particles, i.e., 100 nm PSL particles (shortly, 'PSL') and ammonium sulfate (abbreviated as 'AS') during dry scans (RH < 10 %), and those observed for three types of Indonesian peatland burning particles (i.e., acacia, fern, and peat) following humidification (RH = 90 %), into the supplementary material (as displayed in Fig.S3 below).

It is obvious that the size distributions of selected 100 nm PSL and ammonium sulfate particles are monodispersed and mainly located within the size range of 90 – 120 nm. Size distributions of both peat and acacia burning particles are quite similar to those of PSL and ammonium sulfate particles, with a slightly wider size range of 90 − 130 nm. The fern burning particles tend to be more hygroscopic than peat and acacia burning particles and hence possess a wider size distribution ranging from 100 nm to 140 nm.

*Page 9, Line 12:* *'In all cases, narrow monodisperse distributions were observed (see Figure S3), demonstrating that chemical compositions of particles were uniform (Gysel et al., 2007; Carrico et al., 2010).'*

[Figure]

*Fig. S3* *Normalized particle number size distributions of 100 nm PSL and ammonium sulfate (AS) particles under dry scans (RH < 10%), and of 100 nm peat, acacia, and fern burning particles following humidification (RH = 90%) measured with the HTDMA system.*

**R1C5: pg. 10: It is unclear why Eq. (2) is provided. The data in the Table 2 appear to be calculated from the data. For a mix of compounds, the Eq. (2)**

**should be formulated for multiple components. Furthermore, Eq. (2) is only valid for infinitely soluble compounds. If the equation is used later, the relationship to solubility should be made clearer, especially in the context to the water and octanol extracts.**

**Response**: We showed the equation (2), as it was employed to estimate $\kappa$ from the experimental data of Asa-Awuku et al. (2008). The equation was not directly used for analysis of our data. The corresponding description was updated as follows so that the purpose of the equation will be clearer to the readers.

*Page 10, Line 18: '... temperature (298 K). The calculated $\kappa$ results for our HTDMA measurements are summarized in Figure 3 and Table 2. It is worth noting that $\kappa$ is related with molar volume of water soluble compounds ($M_s/\rho_s$), which is calculated from both the molecular weight ($M_s$) and density ($\rho_s$) by the following equation (Rose et al., 2008):*

$$\kappa = i_s \frac{\rho_s M_w}{\rho_w M_s}, \tag{1}$$

*where i is van't Hoff factor. This equation (2) was mainly employed to derive $\kappa$ from the experimental data of Asa-Awuku et al. (2008), which has calculated the mean molar volume of WSOM extracted from biomass burning particles with a CCN measurement.'*

**R1C6: pg. 11: "the value of $\kappa$ for acacia burning particles is similar to that was measured for WSOM extracted from a prescribed forest fire experiment in Georgia (USA) ($\kappa = 0.10$), which was estimated from a molar volume of $1.6 \times 10^{-4} \, \text{m}^3 \, \text{mol}^{-1}$ (Asa-Awuku et al., 2008)." And "The $\kappa$ value of biomass**

burning WSOM separated by XAD-8 is estimated as 0.29, using molar volume $(6.2 \times 10^{-5} \text{ m}^3 \text{ mol}^{-1})$ estimated from a CCN measurement by Asa-Awuku et al. (2008)." It is unclear what is meant here. Did Asa-Awuku measure water uptake or CCN and compute kappa? Did they measure (average) molecular weight and kappa is calculated from that? If so, what is the relevance? Please explain.

**Response**: Asa-Awuku et al. (2008) measured CCN activity, and estimated average molecular weight assuming surface tension of water and complete solubility. These values were used for the subsequent estimation of corresponding $\kappa$ value. As discussed in Kuwata and Lee (2017), hydrophilic fraction separated by the 1-octanol-water partitioning method (volume ratio of 1-octanol:water = 1:1) has very similar properties as those classified by the XAD-8 column. The purpose of the description was to compare $\kappa$ of hydrophilic faction of biomass burning particles measured by the present and previous studies. Following description was added to clarify this point.

*Page 11, Line 23:* *'...and 0.18 (fern), respectively. Although it is the first hygroscopic measurement for WSOM classified with 1-octanol-water liquid-liquid extraction technique, the value could be compared with those for hydrophilic fractions classified by XAD-8 column.'*

**R1C7: pg. 12: (Lee et al., in preparation). I believe papers in preparation should not be cited in ACP articles.**

**Response:** We have deleted the citation with the whole sentence.

**Reference:** Kuwata, M., and Lee, W.-C. (2017). 1-Octanol-Water Partitioning as a Classifier of Water Soluble Organic Materials: Implication for Solubility Distribution. Aerosol Sci. Technol., 51(5): 602-613.

Kuwata, M., Shao, W., Lebouteiller, R., and Martin, S. T. (2013). Classifying organic materials by oxygen-to-carbon elemental ratio to predict the activation regime of Cloud Condensation Nuclei (CCN). Atmos. Chem. Phys., 13(10): 5309-5324.

Psichoudaki, M., and Pandis, S. N. (2013). Atmospheric Aerosol Water-Soluble Organic Carbon Measurement: A Theoretical Analysis. Environ. Sci. & Technol., 47(17): 9791-9798.

**Anonymous Referee #2**

**General comments:**

**R2C0: This is a very interesting study focusing on the water uptake of organic-dominated biomass burning aerosols in Indonesia. The very low hygroscopicity was attributed to the small fraction of water soluble organic matter. The authors also demonstrated the importance of biomass types in controlling the kappa and water soluble organic matter fraction and the role of highly oxygenated organic compounds in controlling aerosol hygroscopicity. Overall, it is a convincing study that may help improve our understanding on hygroscopicity of aerosol particles, especially organic aerosols. Previous work was properly referred and the paper was well written. I would recommend its publication if the authors could address a couple of minor issues listed below.**

**Response:** We appreciate the reviewer's interest in our work and thanks for the supportive comments.

**Other comments**

**R2C1: Based on correlations between WSOC fraction and kappa, the authors suggested the importance of WSOC fraction in determining kappa of organic aerosol particles. My question is if and how we can use the WSOC fraction and kappa of $A_0$ and $A_1$ aerosols to predict kappa, for example under the Zdanovskii, Stokes, and Robin (ZSR) assumption? Do we need additional information to improve the kappa prediction?**

**Response:** We acknowledge the reviewer for the insightful comment. We agree with the reviewer that $\kappa$ of biomass burning organic aerosol particles can be estimated using WSOC fraction ($f_{WSOC}$) and $\kappa$ of WSOM ($\kappa_{WSOM}$) under some assumptions. These assumptions include thermodynamic approximations such as ZSR and volume additivity as well as relationships between chemical compositions of bulk and individual particles. For instance, we observed size-dependences in $\kappa$ both for fresh burning and $A_0$ particles, suggesting that chemical composition was dependent on diameter. Although quantifications of $f_{WSOC}$ and $\kappa_{WSOM}$ provide useful insights into controlling factors of hygroscopic growth, approximations mentioned above also need to be considered for quantitative analysis. The fraction of $A_1$ in $A_0$ can be estimated using the approach proposed by Kuwata and Lee (2017). Even though separation of $A_1$ is useful in obtaining chemical insights into how water uptake property is regulated, similar approximations still need to be applied for direct application. It is clarified in the revised manuscript as follows.

***Page 15, Line 18:*** *'The present study demonstrates the importance of water-soluble organic fraction, which includes highly soluble one, in quantifying the hygroscopic growth of freshly emitted biomass burning particles. Addition of these different fractions could provide an accurate estimation on hygroscopic growth, which is based on theoretical background. Only size-unresolved bulk chemical data were employed for the present study. However, chemical characteristics of actual atmospheric particles could depend on both particle size and mixing state. These factors would also need to be considered in applying the laboratory data to future studies.'*

**R2C2: Page 3 line 17, can the authors clarify how "hygroscopic" particles or "CCN-active" are defined here? Because a kappa of 0.02 was considered as hygroscopic while a higher kappa of 0.05 was considered as CCN inactive. Concerning the range of kappa in literature, a lower kappa of 0.01 has been reported for low-volatile biomass burning aerosol particles in Rose et al. (2008).**

**Reference: Rose, D., Gunthe, S. S., Su, H., Garland, R. M., Yang, H., Berghof, M., Cheng, Y. F., Wehner, B., Achtert, P., Nowak, A., Wiedensohler, A., Takegawa, N., Kondo, Y., Hu, M., Zhang, Y., Andreae, M. O., and Pöschl, U.: Cloud condensation nuclei in polluted air and biomass burning smoke near the mega-city Guangzhou, China – Part 2: Size-resolved aerosol chemical composition, diurnal cycles, and externally mixed weakly CCN-active soot particles.**

**Response:** We appreciate the reviewer for pointing out this issue. In our original ACPD paper, we defined $\kappa$ values lower than 0.01 as nearly hydrophobic under sub-saturated conditions. On the other hand, the CCN-inactive conclusion in Dusek et al.'s study (2005) was mainly drawn from the experimental result that size-resolved activation ratios of fresh Indonesian peat burning particles were generally lower than 50 % even for the measurements of large (diameter > 150 nm) particles under a high supersaturation of 1.64 %. The corresponding $\kappa$ value of 0.05 for 100 nm particles was subsequently derived from the direct CCN measurements using the $\kappa$-Köhler model with an estimated critical supersaturation and an assumed surface tension equal to that of the air/pure water interface (i.e., 0.072 N m$^{-1}$ at 25 °C). It should be noted that the actual $\kappa$ value could be lower than 0.05 if surface tension of peat burning particles was lower than that for water, as organic particles with a lower surface tension are suggested to be more CCN- active (i.e., corresponding to a lower $\kappa$ value) under a given supersaturation condition (Ovadnevaite et al., 2017). Nevertheless, this $\kappa$ value (0.05) is still within the mean $\kappa$ range (i.e., 0.02−0.06) for the 100 nm Indonesian peat burning particles as what we derived from the HTDMA measurements in this study, thus should be reliable.

In the revised manuscript, we have included the corresponding clarification into the introduction. Relevant details are provided in the discussion of hygroscopicity parameter results calculated from our laboratory measurements in Sect. 3.2.

***Page 3, Line 30:*** *'and they were almost CCN inactive especially for particles larger than 150 nm (equivalent to $\kappa = 0.05$ for 100 nm particles, calculated with an assumed surface tension of 0.072 N m$^{-1}$ at 25 °C) (Dusek et al., 2005).'*

**Reference:** Dusek, U., Frank, G. P., Helas, G., Iinuma, Y., Zeromskiene, K., Gwaze, P., Hennig, T., Massling, A., Schmid, O., Herrmann, H., Wiedensohler, A., Andreae, M. O. (2005). "Missing" cloud condensation nuclei in peat smoke. Geophys. Res. Lett., 32(11): L11802.

Ovadnevaite, J., Zuend, A., Laaksonen, A., Sanchez, K. J., Roberts, G., Ceburnis, D., Decesari, S., Rinaldi, M., Hodas, N., Facchini, M. C., Seinfeld, J. H., and O'Dowd, C. (2017). Surface tension prevails over solute effect in organic-influenced cloud droplet activation. Nature, 546(7660): 637-641.

[revised manuscript text omitted]

NMR samples were prepared by dissolving particles collected on a filter in either $CDCl_3$ or $D_2O$. $CDCl_3$ dissolves most of organic compounds, including both water soluble and insoluble species. On the other hand, only water-soluble organic species will dissolve in $D_2O$ (Decesari et al., 2000; Graham, et al., 2002). The NMR spectra were measured using Bruker AMX-300 spectrometer at 300 MHz frequency.

[Figure]

**Fig.S3 S4** Functional group analyses for peat sampled from a burnt area (Riau-1), acacia and fern burning particles with (a) $CDCl_3$ and (b) $D_2O$, respectively. Four major functional groups identified from [1]H NMR analysis indicate that aliphatic compounds containing the H-C structure are the most abundant in fresh Indonesian peat, fern and acacia burning particles, while oxygenated compounds containing the H-C-C= and H-C-O groups are more likely to dominate in water soluble organic materials (Lee et al., in preparation). An example of the corresponding [1]H NMR spectra for peat burning particles dissolved in $CDCl_3$ can be found in Kuwata et al. (2017). Note that the NMR result of the peat sample in the $D_2O$ case is only qualitative due to very weak signals were detected.

**S4S5.** Correlations between $\kappa$ and OM mass spectra (mainly focusing on $f_{44}$ and $f_{60}$)

[Figure]

**Fig.S4 S5** Correlation of $\kappa$ and $f_{44}$ for all the online and offline measurements.